# Analytical Approach and Numerical Simulation of Reinforced Concrete Beams Strengthened with Different FRCM Systems

**DOI:** 10.3390/ma14081857

**Published:** 2021-04-08

**Authors:** Luis Mercedes, Christian Escrig, Ernest Bernat-Masó, Lluís Gil

**Affiliations:** Department of Strength of Materials and Structures in Engineering, Universitat Politècnica de Catalunya UPC, ESEIAAT, Building TR45, c/ Colom 11, 08222 Terrassa, Spain; christian.escrig@upc.edu (C.E.); ernest.bernat@upc.edu (E.B.-M.); lluis.gil@upc.edu (L.G.)

**Keywords:** concrete beam, bending, cementitious matrix, FRCM, analytical model, numerical model

## Abstract

Fabric-reinforced cementitious matrices (FRCMs) are a novel composite material for strengthening structures. Fabric contributes to tying cross-sections under tensile stress. The complexity of the interfaces between the fabric and the matrix does not allow having a simple and accurate model that enables practitioners to perform feasible calculations. This work developed an analytical approach and a numerical simulation based on the reduction of FRCMs’ strength capabilities under tensile stress states. The concept of effective strength was estimated for different types of fabrics (basalt, carbon, glass, poly p-phenylene benzobisoxazole (PBO), and steel) from experimental evidence. The proposed models calculate the ultimate bending moment for reinforced concrete (RC) structures strengthened with FRCMs. The numerical models performed simulations that reproduced the moment–deflection curves of the different tested beams. Steel fabric showed the highest contribution to strength (78%), while PBO performed the worst (6%). Basalt and carbon showed irregular contributions.

## 1. Introduction

Fabric-reinforced cementitious matrices (FRCMs) have been shown to be one of the most promising retrofitting techniques for reinforced concrete elements [1]. The matrix of this composite material avoids using organic products, which provides FRCMs with a better chemical compatibility with inorganic substrates and diminishes several of the drawbacks of resin-based composite materials [2]. These drawbacks include a lack of vapor permeability, poor fire resistance, non-applicability on wet surfaces or at low temperatures, and chemical hazards for workers who apply strengthening solutions.

Cement-based composites used as a flexural strengthening system for reinforced concrete (RC) structures have been studied experimentally by several authors [3,4,5,6,7,8,9,10,11,12,13,14,15,16,17,18,19]. In all of these studies, FRCMs provided an enhancement of the ultimate flexural capacity of the RC element, demonstrating similar performance to fiber-reinforced polymer (FRP) systems in certain cases. Nevertheless, the use of a cementitious matrix presents new complex failure modes, such as complex slip between the matrix and fibers (see D’Ambrisi and Focacci [3]) and matrix–substrate interactions (see Tommaso et al. [20] and Leone et al. [21]), providing non-optimal development of the ultimate tensile capacity of fibers. Kurtz and Balaguru [4] tested carbon fiber layers, as did Wiberg [12] and Toutanji and Deng [13]. Meanwhile, Barton et al. [14] and Sneed et al. [15] used steel fibers. Brückner et al. [16] included bending and shear. Papanicolaou and Papantoniou [17], Ombres [18], Larrinaga [19], and Babaeidarabad et al. [5] developed particular flexural formulations. Si Larbi et al. [6] analyzed hybrid solutions. Elsanadedy et al. [7] included numerical models to deal with textile reinforcement. Pellegrino and D’Antino [8] studied the reinforcement on prestressed beams, as did Gil et al. [9]. Napoli et al. [10] used steel-reinforced polymer/grout (SRP/SRG) systems, and finally, Ebead et al. [11] developed an effective approach for strengthening beams.

Regarding analytical studies, little research has proposed models for predicting the contribution of FRCMs to the ultimate bending moment. D’Ambrisi and Focacci [3] developed an analytical model based on the evaluation of the effective strain of FRCMs, which is related to the debonding fiber strain. Concurrently, Ombres [18] evaluated the effectiveness of FRP models for predicting the flexural capacity provided by poly p-phenylene benzobisoxazole (PBO)-FRCMs. This investigation concluded that FRP models can predict the capacity of FRCMs in cases in which premature failure modes of the RC elements are avoided. However, these predictions were not accurate when debonding failures occurred. Finally, Babaeidarabad et al. [5] and Ebead et al. [11] assessed the capacity for prediction of the model included in the FRCM code published at the present time (ACI 549.4R-13 [22]) with different results: While the first authors showed that the predicted flexural strengths were conservative with respect to the experimental results, the second authors concluded that the values obtained from the theoretical formulation satisfactorily predicted the mechanical behavior of carbon- and PBO-FRCMs.

Focusing on the numerical models for FRCMs, there are studies such as the one presented by Donnini et al. [23] that used a variational model to obtain information about the mechanical behavior of FRCM composites, particularly at the interface between the tuft and the matrix. The numerical models presented in this study were able to reproduce the behavior of the FRCMs during debonding tests in the case of fabrics constituted by uncoated fibers. This model also allowed establishing an effective joint length of approximately 200 mm, since the maximum load did not increase as the bond length between the mesh and the matrix increased further.

Complementarily, a study by Grande and Milani [24] presented a numerical model dedicated to investigating the influence of different variables on the FRCMs’ strength. This work analyzed the progressive damage of the upper mortar layer (the layer that is not in contact with the element to be strengthened) that affects both the local mechanism of transfer of the shear stresses between the fibers and the upper mortar layer and the overall response of the strengthening system. For this purpose, the authors proposed a simple but effective spring model, where each component of the FRCM system (mortar, fabric, support, and fabric–mortar interface) is modeled along springs with linear or non-linear behavior.

Additionally, one study presented by Sucharda [25] used a nonlinear analysis for detailed failure modeling based on a 3D model and a fracture plastic material model for concrete. This approach made it possible to describe the overall load-bearing capacity, as well as the mechanism of failure and the collapse of the analyzed beams.

Another study, presented by Valikhani et al. [26], designed a numerical simulation to characterize the interfacial properties of concrete substrates and their effect on the bond strength between them and a ultra-high-performance concrete used as a repair material. From all possible modes of failure, the most desirable is that which drives to the breakage of the strengthening material. In this type of failure, the composite material collaborates at its maximum capacity and respects the integrity of the substrate and the bonding material. This paper provides an analytical methodology to calculate the FRCM strengthening systems acting as flexural RC retrofitting. The proposed model introduces a reduction of the tensile capacity of the fibers to define an effective strength in order to represent complex inner phenomena, including slipping or partial fiber breakage inside the matrix. This approach aims to simplify the calculation methods to make the design of FRCM strengthening solutions easier. To fulfil this aim, an experimental campaign consisting of flexural tests on RC beams strengthened with different FRCM systems was conducted by authors [27]. Furthermore, in order to verify the utility of this effective strength, a numerical model was developed. This numerical model was useful to reproduce the experimental moment–deflection curves from the strengthened RC beam. Hence, the proposed novel simplified method of effectiveness strength was doubly validated by comparison with experimental and numerical solutions. 

## 2. Experimental Campaign, Materials, and Methods 

The authors developed an extensive campaign testing 11 full-scale RC beams (see Figure 1) deficient in steel-reinforced flexure, cast with high-strength concrete and flexurally strengthened with different types of cementitious-matrix composite materials. Details of the materials, test setup (four-point flexural test), failures, and results can be found in Escrig et al. [27]. Nevertheless, it is worth noting that strengthening fabrics cover a variety of materials: Basalt (B), carbon (C), glass (G), PBO (P), and steel (S). Additionally, four different additive-modified mortars were used as FRCMs: Mortars P and X1 were designed to be applied to masonry, whereas mortars R and X2 are usually smeared on concrete. Three different batches of concrete were used. In all cases, the concrete was HA/40/F/12/I and the cement used to produce the concrete was CEM 42.5 according to the Spanish concrete designation [28]. The mechanical properties of the concrete and steel data were obtained according to the specifications included in EN 12390-1 [29], EN 12390-3 [30], and EN ISO 15630-1 [31]. The concrete compression strength (fc) of each batch is shown in Table 1, and the tensile strength, yield stress, and Young’s modulus of the steel reinforcement were 634 MPa, 517 MPa, and 198 GPa, respectively.

The compression strength of the cementitious matrix (experimental test) and the mechanical properties (supplied by the manufacturer) of the fabrics necessary to define the material properties of the analytical and numerical models are summarized in Table 1 and Table 2.

The specimens were tested under a four-point flexural test with a free span between supports of 4.00 m in length. The supports were steel cylinders that allowed free rotation in the plane of the beams. The load was applied using a hydraulic actuator of 250 kN, and transferred to the tested specimens through a steel distribution beam with two application points separated 1.40 m. The deflections were obtained using six potentiometers placed symmetrically in pairs on each side of the section. Deflections of the other two analyzed sections, corresponding to the load application points, were controlled by laser position transducers. All data were continuously recorded at a frequency of 50 Hz using a data acquisition system (more information in [27]).

A summary of the experimental results of the bending loading tests are shown in Table 3. This shows the experimental values of the maximum flexural moment (*M_max,exp_*), the yielding flexural moment (*M_y,exp_*), and the deflections recorded at the mid-span of the specimens when the maximum flexural moment (*δ_max,exp_*) and the yielding flexural moment (*δ_y,exp_*) were reached. 

The results presented in Table 3 show coefficients of variation between 0% and 14%, indicating good accuracy and repeatability of the experiments.

## 3. Analytical Model

### 3.1. Considerations

A new approach for a classical analytical method for estimating the ultimate bending moment of RC beams flexurally strengthened with FRCMs is proposed. The increase in the ultimate flexural capacity provided by the strengthening system depends on the capacity of the grids to distribute the stresses uniformly inside the matrix. Papanicolaou and Papantoniou [17] and D’Ambrisi and Focacci [3] listed the multiple aspects that can affect the stress transfer mechanism between the RC beam and the FRCM-strengthening material:In the grid: The type of fiber, the arrangement of the fibers in the yarn (dry fiber fabric or coated yarn grid), and the geometric configuration of the grid.In the matrix: The chemical composition of the mortars and the size of the fine grain.In the RC beam: The type of concrete and the substrate treatment before the application of FRCMs.

It is easy to understand that the complexity of interfaces and the chemical mechanism among yarns, grids, coatings, and mortars that determine the stress distribution in the composite material is a matter that goes far beyond the scope of the present work. It is an extremely complex problem that requires a multi-scale approach with the assessment of chemists and physicists. The following analytical method predicts the flexural ultimate capacity of the strengthened RC beams for each specific strengthening system by reducing the tensile capacity of the FRCM materials using a reduction coefficient (*β*) applied to the ultimate tensile capacity of the fibers. Hence, it approaches the problem from a simplification, reducing the complexity to a single effective fiber strength capacity limited by *β*. This parameter is determined for each type of strengthening fabric by means of the adjustment of the analytical predictions of the maximum flexural moment (*M_max,an_*) with the experimental results (*M_max,exp_*) of the database composed by the tests carried out by the authors and the evidence found in the literature review. The intention of this work was to provide a useful tool to help practitioners to estimate the complex performance from a feasible formulation.

### 3.2. Formulation

The failure of strengthened systems happens at a lower capacity of the theoretical maximal strengthening capacity of a composite. The explanation for this combines different complex inner failures that weaken the capacity of the composite, as stated in the previous paragraph, particularly the relative slip between the fibers and matrix in the contact surfaces, as well as the relative slip between the yarns in contact with mortar and the core yarns and the effect of coating on the fibers. These are properties difficult to estimate during the execution of the reinforcement cast-in-place. Another additional mechanism is the partial breakage of yarns in the fabric. The overall problem is obviously very complex because it depends on the surface properties, penetration of mortar, yarn shapes, boundary contacts, and execution.

Hence, the analytical method for determining the ultimate flexural capacity of the strengthened RC beams is based on the following assumptions: (1) Failure of the strengthening composite while the substrate and bonding maintain their capacities, (2) strain compatibility during the loading process, (3) equilibrium of the forces of the load-bearing cross-section, and (4) reduction of the tensile fiber capacities by a mixed failure of fibers and mortar relative slip. 

The constitutive behavior of concrete, steel, and fibers is shown in Figure 2 according to Eurocode 2 [32]. In the case of concrete, bilinear simplification was considered (Figure 2a). Regarding the steel, an elastic–plastic diagram was used, considering the strain-hardening phenomenon after yielding (Figure 2b). The fibers were assumed to be linear–elastic until failure (Figure 2b). The tensile strength of the concrete and the mortar-FRCM was not considered.

Similar to the analytical approach carried out by Wiberg [12] to calculate the maximum flexural moment (*M_max,an_*), concrete and tensile steel can reach their ultimate capacities in compression and tension according the failure domain. In this case, the ultimate strain of the steel reinforcement (*ε_s,u_*) is considered to be 90‰, and the ultimate strain of the concrete in compression is considered to be 3.5‰. The ultimate tensile capacity of the fibers is reduced by the coefficient *β* to compensate the slipping effect of the yarns inside the matrix. Hence, the contribution of the composite is an effective strength over the total strength of the cross section. Figure 3 shows the internal force equilibrium and the strain distribution of a rectangular RC beam cross-section flexurally strengthened with FRCMs. In this figure, it can be observed that the model considers the different levels of steel reinforcement separately. Furthermore, the compression concrete block is simplified by the assumption that the block works at its maximum capacity (*f_c_*), and its high equals 0.8 times the neutral axis depth (*x*) [32]. 

According to Figure 3, the analytical maximum flexural moment (*M_max,an_*) is calculated as follows (Equation (1)):

Unstrengthened beam:(1)$$M_{cmax,an}\;=\;M_{c,c}+M_{c,s}+M_{c,s,sk}+M_{c,s,2}$$

Strengthened beam:

There was a discrepancy between the analytical ultimate moment ($M_{cmax,an}$) and the experimental ultimate moment ($M_{cmax,\exp}$), probably due to the conservative analysis. The ratio $\frac{M_{cmax,\exp}}{M_{cmax,\mathrm{an}}}$ is 1.26 for this test. Assuming that all of the other concrete beams have the same performance, the strength contribution of the pure RC beam for each one can be modified with this ratio. In Equation (2), it is assumed that this ratio is the same in all of the plastic stages of the strengthened beam. Then, Equation (2) is only useful for the plastic stage from the strengthened beam.
(2)$$M_{cmax,an}\;=\;M_{c,c}+M_{c,s}+M_{c,s,sk}+M_{c,s,2}$$
where

$M_{c,c},M_{s,c}$—the flexural contributions of the concrete (control and strengthened beams);

$M_{c,s},M_{s,s}$—the flexural contributions of tensile steel reinforcement (control and strengthened beams);

$M_{c,s,sk},M_{s,s,sk}$—the flexural contributions of skin steel reinforcement (control and strengthened beams);

$M_{c,s,2},M_{s,s,2}$—the flexural contributions of compressive steel reinforcement (control and strengthened beams);

$M_{sma,an}$—the analytical ultimate maximum strength of the strengthened beam.

The contributions of each withstanding material and neutral axis depth (x) can be determined according to the following equations (Equations (3)–(9)):

Ultimate flexural contributions of the concrete
(3)$$M_{c}\;=\;\frac{0.8f_{c}bx^{2}}{2}$$

Tensile steel reinforcement:(4)$$f_{s,u}\;=\;\varepsilon_{s}E_{s}\;\mathrm{if}\;\varepsilon_{s}<\varepsilon_{s,y}$$
(5)$$f_{s,u}\;=\;\varepsilon_{s}\frac{f_{s,u}-f_{s,y}}{\varepsilon_{s,u}-\varepsilon_{s,y}}+f_{s,y}\;\mathrm{if}\;\varepsilon_{s}\geq \varepsilon_{s,y}$$
(6)$$M_{s}\;=\;A_{s}f_{s,u}\left(d-x\right)$$
where $\varepsilon_{s,y}$ is the elastic limit deformation equal to $\varepsilon_{s,y}\;=\;\frac{f_{s,y}}{E_{s}}$.

Compressive steel reinforcement:

In this case, the same criteria from Equation (6) are fulfilled, and the bending moment is calculated from the following equation:(7)$$M_{s,2}\;=\;f_{s,u}A_{s,2}\left(x-d_{2}\right)$$

Skin steel reinforcement:(8)$$M_{s,sk}\;=\;f_{s,u}A_{s,k}\left(d_{sk}-x\right)$$

Fibers of the strengthening system:(9)$$M_{fib}\;=\;\beta A_{fib}f_{fib,u}\left(d_{fib}-x\right)$$
where the means of all the variables use standard steel–concrete code notation and are depicted in Figure 3. The effectiveness is defined by the reduction coefficient (*β*) introduced in Equation (9).

In the case of this study, fibers that conform have yarns uniformly distributed along the width of the beams. Thus, the area of the fibers (*A_fib_*) is calculated as follows (Equation (10)):(10)$$A_{fib}\;=\;nb_{f}t_{tex}$$
where *n* is the number of grid layers of the FRCMs, *b_f_* is the width of the fabrics, and *t_tex_* is the equivalent thickness of the textile. This last variable is a parameter provided by the corresponding textile manufacturer and represents the thickness of the textile for a continuous distribution of the fibers. In this research, *b = b_f_*.

Finally, to determine the coefficient *β* for each type of fiber, the equality between the analytical predictions of the ultimate bending moment for RC beams flexurally strengthened with FRCMs and the corresponding experimental values is imposed. As a result, an equation of a line in which the reduction coefficient *β* represents its slope is obtained (Equation (11)).
(11)a = βk
where a is the experimental contribution of the FRCMs to the ultimate flexural capacity of the strengthened specimen (Equation (12)), and *k* represents the theoretical contribution of the fibers to the ultimate bending capacity of the beams (Equation (13)):(12)$$a\;=\;M_{su,exp}-\frac{M_{cu,exp}}{M_{cu,an}}\left(M_{c}+M_{s}+M_{s,sk}+M_{s,2}\right)$$
(13)$$k\;=\;A_{fib}f_{fib,u}\left(d_{fib}-x\right)$$

In the case of the FRCM-strengthened concrete beam, it is known that failure of the meshes happens before than the failure of other materials. The ultimate deformation of mesh ($\varepsilon_{f,u}$) is taken to set the point where the maximum load is reached in this case. From the strain compatibility:(14)$$\varepsilon_{c}\;=\;\frac{\varepsilon_{f,u}x}{\;d_{f}-x}$$
(15)$$\varepsilon_{s}\;=\;\frac{\varepsilon_{f,u}\left(d-x\right)}{d_{f}-x}$$
(16)$$\varepsilon_{s,2}=\;\frac{\;\varepsilon_{f,u}\left(x-d_{2}\right)}{d_{f}-x}$$
(17)$$\varepsilon_{s,2}\;=\;\frac{\varepsilon_{f,u}\left(d_{s,k}-x\right)}{d_{f}-x}$$

For the control beam, it is considered that the crushing failure of the concrete occurs before the breakage of the steel. Compressive ultimate deformation of the concrete is taken ($\varepsilon_{c,u}\;=\;0.0035$) to set the point where the maximum tension is reached in this case. Other strains are calculated from this.
(18)$$\varepsilon_{s}\;=\;\frac{\varepsilon_{c,u}\left(d-x\right)}{x}$$
(19)$$\varepsilon_{s,2}\;=\;\frac{\varepsilon_{c,u}\left(x-d_{2}\right)}{x}$$

Once the ultimate strains of the materials are known, one of the following conditions must be fulfilled to determine the maximum moment of the reinforced beam:

$\varepsilon_{c}\leq 0.0035$ (Code [28]).

$\varepsilon_{s}\leq 0.09$ (experimental results [27]).

This analytical model does not consider other failures such as debonding of FRCM strengthening systems. Notice that this case represents the desirable situation for practitioners, in which FRCMs may develop their maximum tensile capacity as a flexural strengthening material. 

### 3.3. Analytical Results

The results obtained from the analytical model are presented in Table 4. This shows the results of the maximum experimental bending moment (*M_max,exp_*) supported by the beams, and the ultimate contributions of the different materials (*ε_c_*, *f_c_*, *f_s,u_*, *f_s,uk_*, and *f_f,u_*), where *f_f,u_* is the tensile stress supported by the fabric after reaching the maximum experimental moment. This was calculated by $f_{f,u}\;=\;\beta f_{fib,u}$. Moreover, Table 4 includes the parameters *a* and *k*, both of which are necessary to obtain the effectiveness coefficient (*β*). 

Table 4 shows that none of the strengthened beams reached the ultimate concrete strain (0.0035) or steel tensile strain (0.09). This means that the FRCM failures occurred before the concrete crushing failure or the steel tensile failure were reached for the strengthened beams. This meets the experimental observations. In the case of the control beam, failure by crushing of the concrete occurred before the breakage of steel.

Additionally, Table 4 shows that the steel-FRCM strengthening systems presented the most efficient behavior with respect to the tensile capacity of the fibers (effective strength of 78%), where the reduction of the flexural capacity provided by the fibers to adapt the mechanical behavior to FRCM systems was less than that of the other fibers. In the case of glass-FRCM, its better efficiency (second, effective strength of 58%) can be explained by the use of a polymer coating of the roving that improves the bonding interface between yarns and mortar. This coating protects the fibers from the breakage caused by friction with the matrix and provides the capacity to distribute the stresses uniformly to the inner and the outer fibers of the yarn (see Voss et al. [33]). This result was also reported by Papanicolaou and Papantoniou [17] in cases in which polymer-coated textiles were used.

However, the FRCMs that presented the highest loss of the withstanding capacity with regard to the tensile strength of the fibers were those made of PBO, carbon, and basalt fabrics. These phenomena may be caused by the poor impregnation of dry fibers by the cementitious matrix, in which the inner filaments of the rovings are not in contact with the mortar, and stress distributions in the yarn are not homogeneous (see Hegger et al. [34]).

The influence of the matrix on obtaining coefficient *β* is noteworthy. This indicates that the proposed analytical methodology is highly sensitive to changes in FRCM components and can be used only when the strengthening solution selected to strengthen the RC structural element is similar to the strengthening solution used for obtaining the reduction coefficient *β*.

## 4. Numerical Models

In order to verify the utility of the *β* coefficients specified in the analytical approach, a numerical model to reproduce the experimental results was developed.

The commercial mechanical simulation software Abaqus^®^ 6.14-4 [35] was used to implement numerical simulations. This choice was based on the aim of using a widely available general purpose simulation tool capable of representing complex material models. In addition, many previous studies based on analysis of FRCMs and reinforced concrete successfully used this software (see, for example, [36,37]).

### 4.1. General Materials’ Constitutive Formulations

The concrete plastic damage model [38] was used to simulate the FRCMs and reinforced concrete. This model is characterized by two elastic moduli: One corresponding to the elastic zone, and another depending on the damage coefficient, which is a function of the cracking situation or the plasticization achieved.

Regarding the plastic zone of the cementitious matrix in tension, it was necessary to define the following parameters (also used in [39]):

Dilatation angle: The first value used for this parameter was 13. It was chosen on the basis of existing literature [37], but convergence difficulties justified increasing this value to 30.

Eccentricity: The predetermined eccentricity suggested by Abaqus was 0.1, which implies that the material has almost the same angle of expansion in a significant range of confining pressure values.

Form parameter of the plasticizing surface (*K*): The default value was equal to 2/3.

Relationship between the maximum uniaxial and biaxial compression stress at the beginning of the loading process (*fb*0/*fc*0). The default value was equal to 1.16.

Viscoplastic regularization: The values of 5 × 10^−6^, 5 × 10^−5^, and 5 × 10^−4^ were tested for an objective choice, proving that the value of 5 × 10^−5^ achieved results that better fit with the experimental results, as well as better model convergence.

Once these material properties were defined, the matrix’s stress–strain curves and the corresponding damage variables were calculated. To calculate these damage variables, the procedure published by [37] was followed. 

### 4.2. Unstrengthened Beam Model

For this analysis, a deformable solid was used to simulate the concrete part defined by a length of 4.4 m, a width of 200 mm, and height of 500 mm (Figure 4a). 

Truss elements were used to simulate steel reinforcement (longitudinal steel bars and stirrups) (Figure 4b). A simplified steel elastic–plastic curve (Figure 2b) obtained from the experimental results was assigned to these reinforcement elements. 

The concrete damage plastic model (previously presented) was used to define the concrete response. The corresponding material properties were obtained by their relationship with the concrete experimental compression strength. These properties refer to the secant modulus of deformation ($E_{ci}$) and the characteristic tensile strength ($f_{ct,k}$) that were calculated from Equations (20)–(22) [28].
(20)$$E_{ci}\;=\;8500\sqrt[{3}]{f_{c}}$$
(21)$$f_{ct,m}\;=\;0.3f_{c}{}^{2/3}$$
(22)$$f_{ct,k\;}=\;0.7f_{ct,m}$$
where $f_{ct,m}$ is the medium tensile strength.

It was necessary to calibrate this property with the aim of reproducing the experimental stiffness and maximum load of the control beam. With this purpose, the modulus of deformation was reduced to 36%, and the 0.7 coefficient from Equation (22) was reduced to 0.35. These modifications were useful to fit the numerical moment–displacement curve with the experimental curve of the control beam.

Two mesh sizes for concrete discretization were tested for convergence analysis: 0.05 m and 0.025 m. The 0.05 m mesh was chosen because no significant difference between the 0.05 mm and 0.025 m meshes was observed (4.7% variation of the maximum reaction force), and the calculation time was 30 times less using the 0.05 m mesh.

Boundary conditions were set according to the four-point flexural setup adopted in the experimental campaign [27]. The displacement in the “y” and “x” directions in one support and the displacement in the “y” and “z” directions in the other were restrained (see Figure 4a) for providing stability to the numerical model. 

The load was directly applied by imposing the vertical displacement that caused the failure of the experimental test of the control beam.

Finally, to identify and check the breaking condition, the steel experimental tensile strength (634 MPa) was considered as the governing criterion. This allowed to reproduce all of the experimental moment–deflection curves.

### 4.3. FRCM-Strengthened Beam Model

In the case of the beams strengthened with FRCMs, the same unstrengthened beam model was used, with the difference that shell elements were added to simulate the FRCMs (see Figure 4c). In the case of the beams strengthened with basalt, glass, and steel FRCMs were used, the corresponding concrete compression strengths (*f_c_*) are presented in Table 4 (different batch).

Shell elements are intended to model structures with one dimension significantly smaller than the other two dimensions. The stresses in the thickness direction have to be negligible to properly use shells. An elastic–plastic model was used on the shell elements to represent the FRCMs. 

To define the mechanical properties of each FRCM (shell elements), it was necessary to implement FRCM models (fabric and mortar) and to determine its tensile behavior with the procedure presented in [40].

This procedure consisted of simulating the FRCMs with a deformable solid (cementitious matrix) and truss elements (fabrics) (see Figure 5), where the fabric was assumed totally bonded (embedded region) to the matrix without allowing sliding in the fabric–matrix interface. To define the material of the cementitious matrix and the fabric, the values presented in Table 1 and Table 2 were used. 

Once the stress–strain curve (see Figure 6) of each type of FRCM was obtained, the modulus of elasticity was obtained from the initial slope, and the stress and plastic strain data of the FRCMs were introduced as the shell elements properties (plastic model). The ultimate stress from the FRCMs imitated when the fabric reached its analytical ultimate stress presented in Table 4 (the *β* coefficients were used).

All of the input material properties used in the FRCM simulation are presented in Table 5.

According to the experimental failures, two types of stress–strain curves were used:

Fabric broke (steel and glass): In these, a discharge slope was defined when the analytical ultimate FRCM stress (*f_fu_*; Table 4) was reached. This was calculated with the same procedure used to calculate the discharge slope in the tensile concrete damage. 

Fabric sliding failures (basalt, carbon, and PBO): In these, a similar discharge slope was defined, but the tensile stress was assumed constant when the discharge slope was proximal to 45% of the analytical ultimate FRCM stress. This is because of the small contribution of the FRCMs during the sliding process of the fabric.

For the interaction between the shell elements (FRCMs) and deformable solids (the concrete beam), a tie connection was used. This approach was considered in other studies [40,41]. A tie connection is a link restriction that allows merging two regions, even though the meshes created on the surfaces of the regions may be different, so complete beam–FRCM bonding can be assumed. 

The same boundary condition to that of the unreinforced beam were imposed. 

All of the input parameters used in this numerical simulation are summarized in Table 6.

### 4.4. Results of the Beams’ Numerical Models

Figure 7 shows the stress contour plots of the concrete beam and the reinforcement steel at the state when the reinforcement steel reached the tensile strength taken as the failure criterion.

The principal stress in the concrete beam shown in Figure 7 describes failure modes similar to the experimental results, with the appearance of flexural cracks and their propagation from the tensile side of the specimens to the neutral axis.

Figure 8 shows the stress contour plots of the different FRCMs at the state when the whole structural element reached the maximum reaction force.

Figure 9 shows the complete moment–deflection curves for all of the beams, including the experimental and numerical results. From experimental values of Figure 9 it can be observed that the greatest contribution to the maximum load from the concrete beam was by steel-FRCM, followed by carbon-FRCM, basalt-FRCM, glass-FRCM, and PBO-FRCM. This order corresponds to the tensile capacity presented in Figure 6. For the case of PBO- and glass-FRCM, the FRCMs significantly decreased the tensile contribution at the middle beam, meaning that the FRCMs reached their maximum stress before the whole structure reached the maximum reaction force.

The results obtained from the numerical model are presented in Table 7. That table shows the results of the maximum flexural moment (*M_max,num_*), the yielding flexural moment (*M_y,num_*), and the deflections recorded at the mid-span of the specimens when the maximum flexural moment (*δ_max,num_*) and the yielding flexural moment (*δ_y,num_*) were reached.

The fitting capabilities of the numerical model were analyzed for all the beams:Control beam: The ultimate and yielding moments and the deflection at the ultimate moment properly fit the experimental results with differences ranging between 2% and 11%. However, the numerically predicted deflection at the yielding moment was much higher than that in the experimental results (39%).Beam strengthened with basalt-, carbon-, glass-, and steel-FRCM: The model for the strengthened beam was able to obtain values of the ultimate moment the and deflection at the ultimate moment close to the experimental results with differences ranging from 2% to 19%, except in the case of glass-FRCM, where the numerical model brought a deflection at the maximum moment with a far lower value than that in the experimental tests (49%).Beam strengthened with PBO-FRCM: This beam showed the greatest dispersion of the deflection at the ultimate moment, the yielding moment, and the deflection with differences ranging between 29% and 97%. This is because of the premature experimental failure of the beam strengthened with PBO-FRCM.

Figure 9 demonstrates that the proposed numerical models are able to reproduce the experimental response with sufficient approximation.

The results of the numerical simulations validate the effectiveness of using the *β* coefficients to determine the ultimate effective contribution of the FRCM systems to RC beams, where the interaction between materials is similar to that considered in this study.

## 5. Conclusions

This work presented an analytical approach and a numerical model for determining the ultimate bending capacity of RC structures flexurally strengthened with FRCMs using an effective strength. These models are based on the lowering of the tensile capacity of the fibers using a reduction parameter, *β*.
Comparing the different types of reinforcements, the steel and glass strengthening systems are the FRCMs that developed the lowest reduction of the tensile capacity of the fibers. Hence, they were the most efficient. However, carbon, basalt, and PBO, very promising materials, showed the highest reduction of tensile capacity of the fibers probably due to the incapacity of the corresponding matrices to impregnate fully the dry fibers that compose the fabric.The analysis of FRCM systems revealed that the type of matrix used highly affects the reduction parameter of fibers, implying that it is necessary to create a database of experimental results for each combination of matrix and grid used. Thus, the use of combinations of grids and matrices to manufacture FRCMs not guaranteed by the provider should be taken with extreme care.The concrete tensile strength and the modulus of deformation were adapted to fit the numerical simulations with the experimental results. The tensile strength coefficient was reduced a 50% (from 0.7 of the normative to a 0.35). The modulus of deformation was reduced to 36%. These values were successfully used for both the control and FRCM-strengthened beams. For all of them, numerical simulations proved that these modified properties were representative of the proposed model.The numerical model was effective for carbon-, basalt-, glass-, and steel-FRCM-strengthened beams for predicting the bending moments and displacement (variation between 2% and 15%) due to the coefficient *β* determined in the analytical approach. However, in the case of PBO-FRMC, it was not possible to reproduce the experimental results, because the failure criterion was reached at a very large deflection bigger (variation of 97%) than its experimental failure.In summary, it can be said that although there is limited experimental evidence to determine the coefficient *β*, it can be successfully used as a representative parameter of the performance capability of different types of FCRM solutions. *β* can be applied as a reduction of the ultimate tensile strength of fibers and it represents a promising approach that highly simplifies the analytical design of FRCMs when applied for RC flexural strengthening. Practitioners could benefit from using this strategy to calculate feasible strengthening solutions. With the aim of increasing the reliability of the model, we urge the performance of more experimental tests that allow to expand the results database and fitting the coefficient *β* more accurately.

## Figures and Tables

**Figure 1 materials-14-01857-f001:**
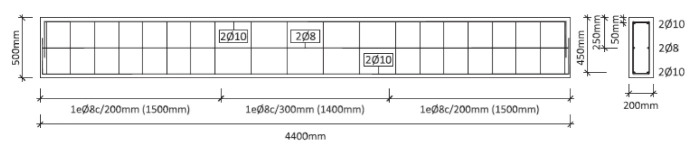
Geometry and steel reinforcement of the beams.

**Figure 2 materials-14-01857-f002:**
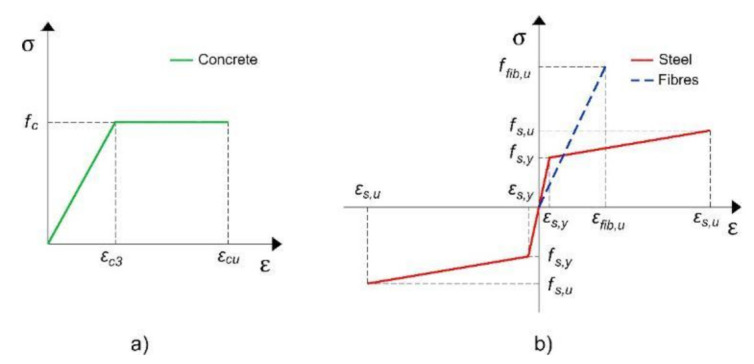
Constitutive behavior of the materials: (**a**) Concrete and (**b**) steel and fibres.

**Figure 3 materials-14-01857-f003:**
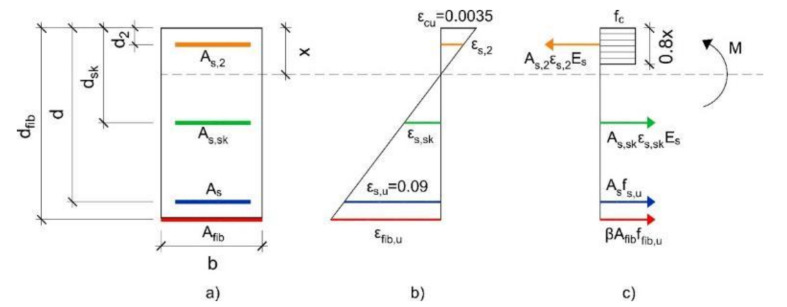
Analysis of the cross-section for the ultimate limit state in bending: (**a**) Geometry, (**b**) strain distribution, and (**c**) force equilibrium.

**Figure 4 materials-14-01857-f004:**
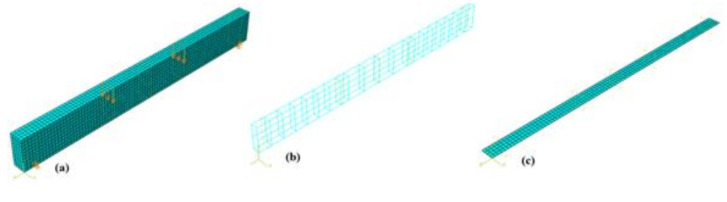
Numerical model of beams: (**a**) Deformable solid beam, (**b**) truss element-reinforced steel, and (**c**) shell element FRCMs.

**Figure 5 materials-14-01857-f005:**
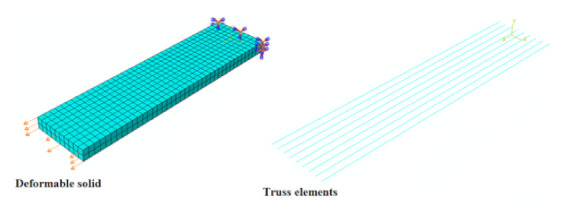
FRCM models.

**Figure 6 materials-14-01857-f006:**
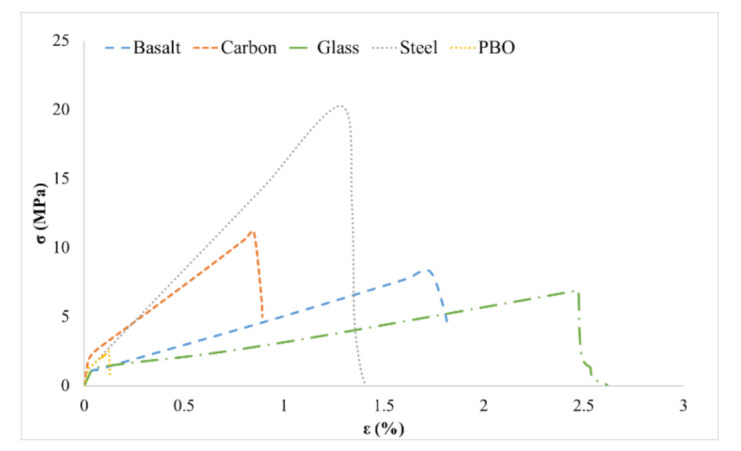
FRCM stress–strain diagrams (numerical model results).

**Figure 7 materials-14-01857-f007:**
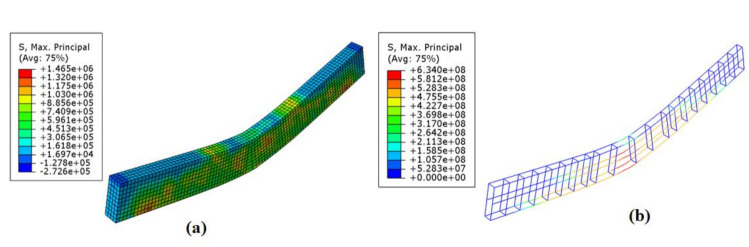
Max. principal stress (N/m^2^) state of the simulation beam: (**a**) Concrete beam and (**b**) steel reinforcement.

**Figure 8 materials-14-01857-f008:**
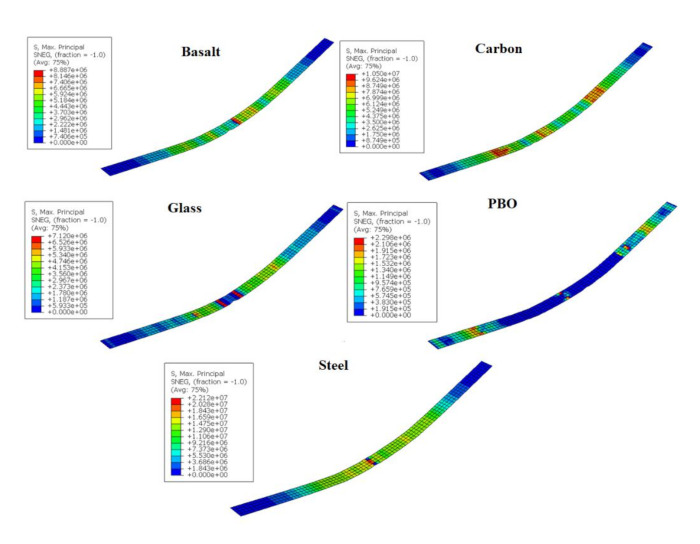
Max. principal stress (N/m^2^) state of the FRCMs.

**Figure 9 materials-14-01857-f009:**
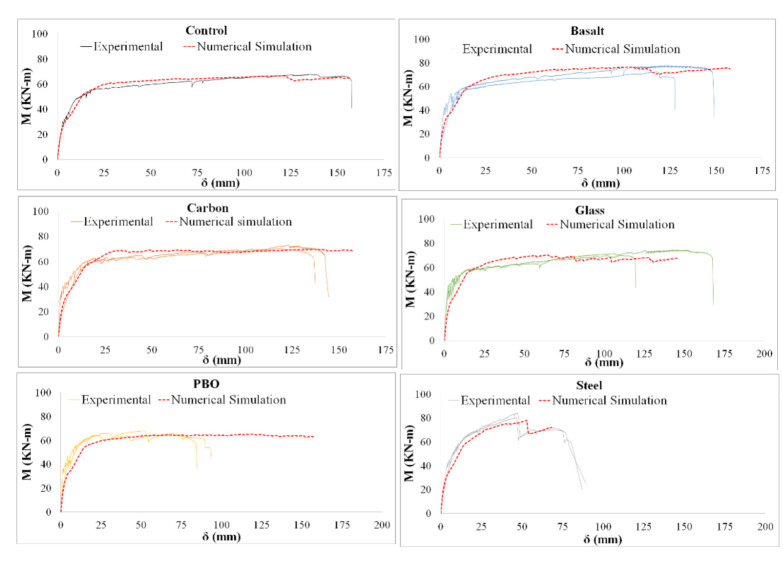
Moment–deflection diagrams.

**Table 1 materials-14-01857-t001:** Concrete and mortar properties.

Beam State	Concrete Compression Strength (MPa)		Mortar-FRCM Compression Strength (MPa)	
Control beam	42.35	(3%)	-	-
Beam with basalt-FRCM	55.42	(1%)	24.65	(6%)
Beam with carbon-FRCM	42.35	(3%)	24.95	(7%)
Beam with glass-FRCM	46.52	(3%)	35.4	(7%)
Beam with PBO-FRCM	42.35	(3%)	30.02	(7%)
Beam with steel-FRCM	46.52	(3%)	24.65	(6%)

(%) Coefficient of variation. FRCM, Fabric-reinforced cementitious matrix; PBO, poly p-phenylene benzobisoxazole.

**Table 2 materials-14-01857-t002:** Fabrics and cementitious matrix properties.

FRCM Properties	Basalt	Carbon	Glass	PBO	Steel
Fabric tensile strength (MPa)	3080	4320	2610	5800	3200
Fabric modulus of elasticity (MPa)	95,000	240,000	90,000	270,000	206,000
Fabric area (mm^2^)	10.6	9.4	8.4	9.1	15

**Table 3 materials-14-01857-t003:** Experimental results of the bending tests.

Beam	No. of Test	*M_max,exp_* (kN-m)	*M_y,exp_* (kN-m)	*δ_max,exp_* (mm)	*δ_y,exp_* (mm)
Control	1	67.89	48.66	135.08	10.05
Basalt	2	75.05	58.62	120.78	15.10
		(4%)	(1%)	(2%)	(4%)
Carbon	2	71.62	60.12	120.44	16.30
		(2%)	(1%)	(2%)	(%)
Glass	2	72.35	58.04	128.18	14.54
		(3%)	(%)	(14%)	(9%)
PBO	2	66.26	63.56	60.35	20.39
		(3%)	(1%)	(16%)	(8%)
Steel	2	82.10	69.64	46.47	22.82
		(2%)	(1%)	(0%)	(8%)

(%) Coefficient of variation.

**Table 4 materials-14-01857-t004:** Analytical model results.

Beam	*M_u,exp_* (kN-m)	*ε_c_* (/)	*f_c_* (MPa)	*f_s,u_* (MPa)	*f_s,uk_* (MPa)	*a* (kN-m)	*K* (kN-m)	*β*	*f_f,u_* (MPa)
**Control**	67.89	0.0035	42.35	579.634	549.85	14.12	-	-	-
**Basalt**	75.05	0.00173	55.42	500	520.00	8.43	15.65	0.54	1658.32
**Carbon**	71.65	0.00241	42.35	538.375	528.25	8.24	19.30	0.43	1844.09
**Glass**	72.40	0.00169	46.52	551.456	535.21	6.06	10.46	0.58	1512.73
**PBO**	66.25	0.00146	42.35	542.493	530.35	1.42	24.96	0.06	329.41
**Steel**	82.10	0.00091	46.52	535.455	526.75	17.92	19.30	0.78	2504.40

**Table 5 materials-14-01857-t005:** Input material properties to the FRCM simulation.

*FRCM*	Mortar—Young’s Modulus (MPa)	Mortar—Compression Strength (MPa)	Mortar—Tension Strength (MPa)	Fabric-Young’s Modulus (MPa)	Fabric—Tension Strength × *β* (MPa)
Basalt	8905.54	24.65	0.89	95,000	1658.32
Carbon	8941.53	24.95	0.90	240,000	2048.99
Glass	10,047.45	35.40	1.13	61,250	1512.73
PBO	9510.24	30.02	1.01	270,000	329.41
Steel	8905.54	24.65	0.89	206,000	2504.40

**Table 6 materials-14-01857-t006:** Input materials properties to simulate unstrengthened and strengthened beams.

Concrete Beams	Concrete—Young’s Modulus (MPa)	Concrete—Compression Strength (MPa)	Concrete—Tension Strength (MPa)	FRCM—Young’s Modulus (First Slope) (MPa)	FRCM—Tension Strength × *β* (MPa)
Control	10,666.11	42.35	1.28	-	-
Basalt	11,666.58	55.42	1.53	1760.38	7.91
Carbon	10,666.11	42.35	1.28	6187.00	9.63
Glass	11,005.29	46.52	1.36	2438.09	6.35
PBO	10,666.11	42.35	1.28	3518.83	1.50
Steel	11,005.29	46.52	1.36	3887.05	18.60

**Table 7 materials-14-01857-t007:** Numerical simulation results.

Beam	*M_max,num_* (kN-m)	*M_y,num_* (kN-m)	*δ_max,num_* (mm)	*δ_y,num_* (mm)
Control	66.60	52.06	120.76	13.99
Δ_exp_	(−2%)	(7%)	(−11%)	(39%)
Basalt	76.51	57.27	104.01	13.62
Δ_exp_	(2%)	(−2%)	(−14%)	(−10%)
Carbon	69.78	55.81	138.65	14.18
Δ_exp_	(−3%)	(−7%)	(15%)	(−13%)
Glass	70.30	54.99	64.71	13.77
Δ_exp_	(−3%)	(−5%)	(−49%)	(−5%)
PBO	65.30	82.10	119.08	14.02
Δ_exp_	(−1%)	(29%)	(97%)	(−31%)
Steel	77.93	56.44	53.03	13.62
Δ_exp_	(−5%)	(−19%)	(14%)	(−40%)

## Data Availability

Data is contained within the article.

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
