# Peer review of "Analytical Approach and Numerical Simulation of Reinforced Concrete Beams Strengthened with Different FRCM Systems"

_materials, 2021, doi:10.3390/ma14081857_

Round 1

Reviewer 1 Report

This manuscript deals with the analytical approach and numerical simulation of reinforced concrete beams.

This is an interesting topic and research.

The manuscript builds on the authors' previous research.

Interesting experiments of the authors from the publication are used for numerical modeling:
C. Escrig, L. Gil, and E. Bernat-Maso, “Experimental comparison of reinforced concrete beams strengthened against bending 607
with different types of cementitious-matrix composite materials,” Constr. Build. Mater., vol. 137, pp. 317–329, 2017, doi: 608
10.1016/j.conbuildmat.2017.01.106.

The topic is interestingly processed, but some information is already known.

The chosen approach to numerical modeling is possible. It is necessary to state in more detail the boundary conditions of the model and the parameters of the solver. 

It is necessary to detail all input parameters for the calculation!!!
It is also necessary to list the interfaces used. 
Without this data, numerical modeling cannot be considered plausible. 
It is necessary to specify the procedure for determining specific input parameters for material models in the computer program. Has any inverse analysis or recommendation approach been used?
I also recommend reworking the chapter introduction and dealing more with the issue of numerical modeling of reinforced concrete structures, recommendations, etc.

Valikhani, A. et. al. Numerical Modeling of Concrete-to-UHPC Bond Strength. Materials 2020, 13, 1379.
Sucharda, O. Identification of Fracture Mechanic Properties of Concrete and Analysis of Shear Capacity of Reinforced Concrete Beams without Transverse Reinforcement. Materials 2020, 13, 2788.

The topic itself is solved logically.

line 171 (Error! Reference source not found.)
line 174 (Error! Reference source not found.)
line 196 (Error! Reference source not found.)
The reference to Table 3 is on page 3. The table is on page 8. Cluttered.

All mechanical properties must be listed in the tables! Mechanical properties from laboratory experiments must be clear !!!
Please table for concrete separately, for reinforcing steel separately - test results (add - standard deviation, VoC) and input parameters used for the material model for the calculation.
Variable parameters are italic! pay attention to tables and text!

Make a list of all used abbreviations and variables - fc, fs, u, ff, u ....... Make a special chapter at the end of the manuscript.

The addressed topic and results must be presented in the context of the current state. 

The manuscript must present a comprehensive approach to this issue.
The authors have done a good job in research, but the informative value of the manuscript must be increased, especially in the discussion and conclusion. 
New findings and benefits of research on why to read the article must be clearly stated.

The manuscript must be revised before publication.

Author Response

It is necessary to state in more detail the boundary conditions of the model and the parameters of the solver.

It is necessary to detail all input parameters for the calculation

The detail of the boundary conditions has been added in line 130-138, and all input parameter for the calculation has been added in table 5,6 and 7

It is also necessary to list the interfaces used.

The interfaces interaction between the materials used in the numerical model are explain in line 414-416, 447-451.

It is necessary to specify the procedure for determining specific input parameters for material models in the computer program. Has any inverse analysis or recommendation approach been used?

The procedure for determining some parameters was referenced in line 390-391. Reference [31].

I also recommend reworking the chapter introduction and dealing more with the issue of numerical modeling of reinforced concrete structures, recommendations, etc.

Valikhani, A. et. al. Numerical Modeling of Concrete-to-UHPC Bond Strength. Materials 2020, 13, 1379.

Sucharda, O. Identification of Fracture Mechanic Properties of Concrete and Analysis of Shear Capacity of Reinforced Concrete Beams without Transverse Reinforcement. Materials 2020, 13, 2788.

The introduction has been modified and suggested references have been added in line 79-86

line 171 (Error! Reference source not found.)

line 174 (Error! Reference source not found.)

line 196 (Error! Reference source not found.)

The reference to Table 3 is on page 3. The table is on page 8. Cluttered.

All these errors have been corrected.

All mechanical properties must be listed in the tables! Mechanical properties from laboratory experiments must be clear !!!

Please table for concrete separately, for reinforcing steel separately - test results (add - standard deviation, VoC) and input parameters used for the material model for the calculation.

Reviewer was right. It has been added the table 1, 5, 6 y 7

Variable parameters are italic! pay attention to tables and text!

This has been corrected.

Make a list of all used abbreviations and variables - fc, fs, u, ff, u ....... Make a special chapter at the end of the manuscript

This has been added at the end of the manuscript, before of the references

The manuscript must present a comprehensive approach to this issue.

The authors have done a good job in research, but the informative value of the manuscript must be increased, especially in the discussion and conclusion.

New findings and benefits of research on why to read the article must be clearly stated.

We add commentaries in lines 89-83, 97-99, and a note in conclusions about practioners’ benefit of the approach.

Reviewer 2 Report

Paper is very interesting, but:
Do you tested all nets? Or data are from technical cards?
Add some work of: DiTommaso, Corradi, Valluzzi,
Please provide more detailed reasoning behind the behaviors. The details should include rigid numbers or percentages.
Please indicate how many samples for each experiment have been used.
Please describe the process of each experiment. Also indicate the model of each tool that is used in the experiment. What is the accuracy of each test? Please explain them accurately.
Figures 1, 3, 4, 8, 9, 10 - please check text size. Is the text in these figures readable?. It is necessary to do this figure again.
Reformat tables 1, 2, 3, 4. 
The manuscript is needed to be revised grammatically. The authors are required to check the whole manuscript with a grammar specialist as it has several grammatical errors.
Please check the format of all references is in the journal template. There are many that do not adjust with the journal indications.

Author Response

Do you tested all nets? Or data are from technical cards?

No, we do not. This article used values from the technical data sheets.

Add some work of: DiTommaso, Corradi, Valluzzi,

Some of this work has been referenced in line 38.

Please provide more detailed reasoning behind the behaviors. The details should include rigid numbers or percentages.

We add some rigid numbers and percentages in line 312, 317 and 320. Also, the work includes the percentage of performance of each reinforcement compared to the control beam.

Please indicate how many samples for each experiment have been used

This was indicated in table 3.

Please describe the process of each experiment. Also indicate the model of each tool that is used in the experiment. What is the accuracy of each test? Please explain them accurately.

the process of each experiment has been added in line 130-138 (more details in reference [27]), and the accuracy was in table 3, has been added a commentary in line 145-146

Figures 1, 3, 4, 8, 9, 10 - please check text size. Is the text in these figures readable?. It is necessary to do this figure again.

Reformat tables 1, 2, 3, 4.

This has been corrected

The manuscript is needed to be revised grammatically. The authors are required to check the whole manuscript with a grammar specialist as it has several grammatical errors.

This has been checked.

Please check the format of all references is in the journal template. There are many that do not adjust with the journal indications.

This has been checked

Round 2

Reviewer 1 Report

Thank you for the adjustments made.
The changes made the improvement of the manuscript.

The research area and results are from the context of the manuscript can better understand.

However, the manuscript can be further edited. 
The information value of the manuscript can be improved.

The manuscript contains the main information.

The manuscript can be published in the journal.

Author Response

Thanks for your suggestions

The manuscript has been revised by a service English languages editing.

Reviewer 2 Report

Thank you for the corrections. Please revise the list of publications cited.

Author Response

Thank you,

This has been revised.

Round 3

Reviewer 2 Report

Paper can be accepted.